# DM-NeRF: 3D Scene Geometry Decomposition and Manipulation from 2D Images

**Bing Wang**[1,2,3]†    **Lu Chen**[1,2]†    **Bo Yang**[1,2]*
[1] Shenzhen Research Institute, The Hong Kong Polytechnic University
[2] vLAR Group, The Hong Kong Polytechnic University    [3]University of Oxford
bingwang@polyu.edu.hk   bo.yang@polyu.edu.hk

## Abstract

In this paper, we study the problem of 3D scene geometry decomposition and manipulation from 2D views. By leveraging the recent implicit neural representation techniques, particularly the appealing neural radiance fields, we introduce an object field component to learn unique codes for all individual objects in 3D space only from 2D supervision. The key to this component is multiple carefully designed loss functions to enable every 3D point, especially in non-occupied space, to be effectively optimized without 3D labels. In addition, we introduce an inverse query algorithm to freely manipulate any specified 3D object shape in the learned scene representation. Notably, our manipulation algorithm can explicitly tackle key issues such as object collisions and visual occlusions. Our method, called DM-NeRF, is among the first to simultaneously reconstruct, decompose, manipulate and render complex 3D scenes in a single pipeline. Extensive experiments on three datasets clearly show that our method can accurately decompose all 3D objects from 2D views, allowing any interested object to be freely manipulated in 3D space such as translation, rotation, size adjustment, and deformation.

## 1 Introduction

In many cutting-edge applications such as mixed reality on mobile devices, users may desire to virtually manipulate objects in 3D scenes, such as moving a chair or making a flying broomstick in a 3D room. This would allow users to easily edit real scenes at fingertips and view objects from new perspectives. However, this is particularly challenging as it involves 3D scene reconstruction, decomposition, manipulation, and photorealistic rendering in a single framework (Savva et al., 2019).

A traditional pipeline firstly reconstructs explicit 3D structures such as point clouds or polygonal meshes using SfM/SLAM techniques (Ozyesil et al., 2017; Cadena et al., 2016), and then identifies 3D objects followed by manual editing. However, these explicit 3D representations inherently discretize continuous surfaces, and changing the shapes often requires additional repair procedures such as remeshing (Alliez et al., 2002). Such discretized and manipulated 3D structures can hardly retain geometry and appearance details, resulting in the generated novel views to be unappealing. Given this, it is worthwhile to design a new pipeline which can recover continuous 3D scene geometry only from 2D views and enable object decomposition and manipulation.

Recently, implicit representations, especially NeRF (Mildenhall et al., 2020), emerge as a promising tool to represent continuous 3D geometries from images. A series of succeeding methods (Boss et al., 2021; Chen et al., 2021; Zhang et al., 2021c) are rapidly developed to decouple lighting factors from structures, allowing free edits of illumination and materials. However, they fail to decompose 3D scene geometries into individual objects. Therefore, it is hard to manipulate individual object shapes in complex scenes. Recent works (Stelzner et al., 2021; Zhang et al., 2021b; Kania et al., 2022; Yuan et al., 2022; Tschernezki et al., 2022; Kobayashi et al., 2022; Kim et al., 2022; Benaim et al., 2022; Ren et al., 2022) have started to learn disentangled shape representations for potential geometry manipulation. However, they either focus on synthetic scenes or simple model segmentation, and can hardly extend to real-world 3D scenes with dozens of objects.

---

*Corresponding Author    † Equal Contribution

Figure 1: The general workflow of DM-NeRF. NeRF (green block) is used as the backbone. We propose the object field and manipulation components as illustrated by blue and orange blocks.

In this paper, we aim to simultaneously recover continuous 3D scene geometry, segment all individual objects in 3D space, and support flexible object shape manipulation such as translation, rotation, size adjustment and deformation. In addition, the edited 3D scenes can be also rendered from novel views. However, this task is extremely challenging as it requires: 1) an object decomposition approach amenable to continuous and implicit 3D fields, without relying on any 3D labels for supervision due to the infeasibility of collecting labels in continuous 3D space; 2) an object manipulation method agreeable to the learned implicit and decomposed fields, with an ability to clearly address visual occlusions inevitably caused by manipulation.

To tackle these challenges, we design a simple pipeline, **DM-NeRF**, which is built on the successful NeRF, but able to **d**ecompose the entire 3D space into object fields and freely **m**anipulate their geometries for realistic novel view rendering. As shown in Figure 1, it consists of 3 major components: 1) the existing radiance field to learn volume density and appearance for every 3D point in space; 2) the object field which learns a unique object code for every 3D point; 3) the object manipulator that directly edits the shape of any specified object and automatically tackles visual occlusions.

The **object field** is the core of DM-NeRF. This component aims to predict a one-hot vector, *i.e.,* object code, for every 3D point in the entire scene space. However, learning such code involves critical issues: 1) there are no ground truth 3D object codes available for full supervision; 2) the number of total objects is variable and there is no fixed order for objects; 3) the non-occupied (empty) 3D space must be taken into account, but there are no labels for supervision as well. In Section 3.1, we show that our object field together with multiple carefully designed loss functions can address them properly, under the supervision of color images with 2D object masks only.

Once the object field is well learned, our **object manipulator** aims to directly edit the geometry and render novel views when specifying the target objects, viewing angels, and manipulation settings. A naïve method is to obtain explicit 3D structures followed by manual editing and rendering, so that any shape occlusion and collision can be explicitly addressed. However, it is extremely inefficient to evaluate dense 3D points from implicit fields. To this end, as detailed in Section 3.2, we introduce a lightweight inverse query algorithm to automatically edit the scene geometry.

Overall, our pipeline can simultaneously recover 3D scene geometry, decompose and manipulate object instances only from 2D images. Extensive experiments on multiple datasets demonstrate that our method can precisely segment all 3D objects and effectively edit 3D scene geometry, without sacrificing high fidelity of novel view rendering. Our key contributions are:

- We propose an object field to directly learn a unique code for each object in 3D space only from 2D images, showing remarkable robustness and accuracy over the commonly-used individual image based segmentation methods.
- We propose an inverse query algorithm to effectively edit specified object shapes, while generating realistic scene images from novel views.
- We demonstrate superior performance for 3D decomposition and manipulation, and also contribute the first synthetic dataset for quantitative evaluation of 3D scene editing. Our code and dataset are available at `https://github.com/vLAR-group/DM-NeRF`

We note that recent works ObjectNeRF (Yang et al., 2021), NSG (Ost et al., 2021) and ObjectSDF (Wu et al., 2022) address the similar task as ours. However, ObjectNeRF only decomposes a foreground object, NSG focuses on decomposing dynamic objects, and ObjectSDF only uses semantic label as regularization. None of them directly learns to segment multiple 3D objects as ours. A few works (Norman et al., 2022; Kundu et al., 2022; Fu et al., 2022) tackle panoptic segmentation in radiance fields. However, they fundamentally segment objects in 2D images followed by learning a separate radiance field for each object. By comparison, our method learns to directly segment all objects in the 3D scene radiance space, and it demonstrates superior accuracy and robustness than

2D object segmentation methods such as MaskRCNN (He et al., 2017) and Swin Transformer (Liu et al., 2021b), especially when 2D object labels are noisy during training, as detailed in Section 4.

## 2 RELATED WORK

**Explicit 3D Representations:** To represent 3D geometry of objects and scenes, voxel grids (Choy et al., 2016), octree (Tatarchenko et al., 2017), meshes (Kato et al., 2018; Groueix et al., 2018), point clouds (Fan et al., 2017) and shape primitives (Zou et al., 2017) are widely used. Although impressive progress has been achieved in shape reconstruction (Yang et al., 2019a; Xie et al., 2019), completion (Song et al., 2017), generation (Lin et al., 2018), and scene understanding (Tulsiani et al., 2018; Gkioxari et al., 2019), the quality of these representations are inherently limited by the spatial resolution and memory footprint. Therefore, they are hard to represent complex 3D scenes.

**Implicit 3D Representations:** To overcome the discretization issue of explicit representations, coordinate based MLPs have been recently proposed to learn implicit functions to represent continuous 3D shapes. These implicit representations can be generally categorized as: 1) signed distance fields (Park et al., 2019), 2) occupancy fields (Mescheder et al., 2019), 3) unsigned distance fields (Chibane et al., 2020; Wang et al., 2022), 4) radiance fields (Mildenhall et al., 2020), and 5) hybrid fields (Wang et al., 2021). Among them, both occupancy fields and signed distance fields can only recover closed 3D shapes, and are hard to represent open geometries. These representations have been extensively studied for novel view synthesis (Niemeyer et al., 2020; Trevithick & Yang, 2021) and 3D scene understanding (Zhang et al., 2021a; Zhi et al., 2021a;b). Thanks to the powerful representation, impressive results have been achieved, especially from the neural radiance fields and its succeeding methods. In this paper, we also leverage the success of implicit representations, particularly NeRF, to recover the geometry and appearance of 3D scenes from 2D images.

**3D Object Segmentation:** To identify 3D objects from complex scenes, existing methods generally include 1) image based 3D object detection (Mousavian et al., 2017), 2) 3D voxel based detection methods (Zhou & Tuzel, 2018) and 3) 3D point cloud based object segmentation methods (Yang et al., 2019b). Given large-scale datasets with full 3D object annotations, these approaches have achieved excellent object segmentation accuracy. However, they are particularly designed to process explicit and discrete 3D geometries. Therefore, they are unable to segment continuous and fine-grained shapes, and fail to support geometry manipulation and realistic rendering. With the fast development of implicit representation, it is desirable to learn object segmentation for implicit surfaces. To the best of our knowledge, this paper is among the first to segment all 3D objects of implicit representations for complex scenes, only with color images and 2D object labels for supervision.

**3D Scene Editing:** Existing methods of editing 3D scenes from images can be categorized as 1) appearance editing and 2) shape editing. A majority of works (Sengupta et al., 2019; Boss et al., 2021; Zhang et al., 2021c; Chen et al., 2021) focus on lighting decomposition for appearance editing. Although achieving appealing results, they cannot separately manipulate individual objects. A number of recent works (Munkberg et al., 2021; Liu et al., 2021a; Jang & Agapito, 2021; Stelzner et al., 2021; Guandao Yang et al., 2021) start to learn disentangled shape representations for potential geometry manipulation. However, they can only deal with single objects or simple scenes, without being able to learn unique object codes for precise shape manipulation and novel view rendering. In addition, there are also a plethora of works (Tewari et al., 2020; Niemeyer & Geiger, 2021; Dhamo et al., 2021; Alaluf et al., 2022) on generation based scene editing. Although they can manipulate the synthesized objects and scenes, they cannot discover and edit objects from real-world images.

## 3 DM-NERF

Given a set of $L$ images for a static scene with known camera poses and intrinsics $\{(\mathcal{I}_1, \boldsymbol{\xi}_1, \boldsymbol{K}_1) \cdots (\mathcal{I}_L, \boldsymbol{\xi}_L, \boldsymbol{K}_L)\}$, NeRF uses simple MLPs to learn the continuous 3D scene geometry and appearance. In particular, it takes 5D vectors of query point coordinates $\boldsymbol{p} = (x, y, z)$ and viewing directions $\boldsymbol{v} = (\theta, \phi)$ as input, and predicts the volume density $\sigma$ and color $\boldsymbol{c} = (r, g, b)$ for point $\boldsymbol{p}$. In our pipeline, we leverage this vanilla NeRF as the backbone to learn continuous scene representations, although other NeRF variants can also be used. Our method aims to decompose all individual 3D objects, and freely manipulate any object in the 3D scene space. To achieve this, we design an object field component to parallelly learn an object code for every query point $\boldsymbol{p}$, together with an object manipulator to edit the learned radiance fields and object fields.

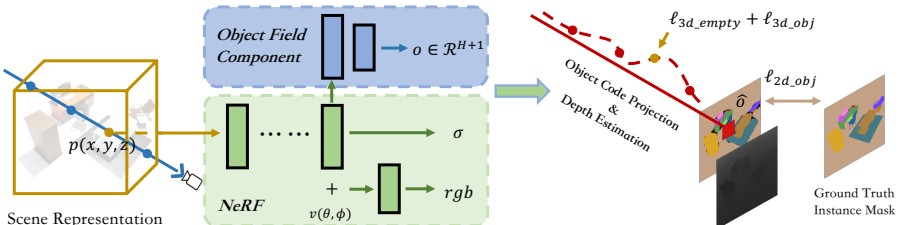

Figure 2: The architecture of our pipeline. Given a 3D point $p$, we learn an object code through three loss functions: $\ell_{2d\_obj}/\ell_{3d\_empty}/\ell_{3d\_obj}$ in Eqs 5&8&9, using 2D and 3D supervision signals.

## 3.1 OBJECT FIELDS

**Object Field Representation:** As shown in Figure 2, given the input point $p$, we model the object field as a function of its coordinates, because the object signature of a 3D point is irrelevant to the viewing angles. The object field is represented by a one-hot vector $o$. Basically, this one-hot object code aims to accurately describe the object ownership of any point in 3D space.

However, there are two issues here: 1) the total number of objects in 3D scenes is variable and it can be 1 or many; 2) the entire 3D space has a large non-occupied volume in addition to solid objects. To tackle these issues, we define the object code $o$ as $H + 1$ dimensional, where $H$ is a predefined number of solid objects that the network is expected to predict in maximum. We can safely choose a relative large value for $H$ in practice. The last dimension of $o$ is particularly reserved to represent the non-occupied space. Notably, this dedicated design is crucial for tackling occlusion and collision during object manipulation discussed in Section 3.2. Formally, the object field is defined as:

$$o = f(p), \quad \text{where } o \in \mathcal{R}^{H+1} \tag{1}$$

The function $f$ is parameterized by a series of MLPs. If the last dimension of code $o$ is 1, it represents the input point $p$ is non-occupied or the point is empty.

**Object Code Projection:** Considering that it is infeasible to collect object code labels in continuous 3D space for full supervision while it is fairly easy and low-cost to collect object labels on 2D images, we aim to project the object codes along the query light ray back to a 2D pixel. Since the volume density $\sigma$ learned by the backbone NeRF represents the geometry distribution, we simply approximate the projected object code of a pixel $\hat{o}$ using the sampling strategy and volume rendering formulation of NeRF. Formally, it is defined as:

$$\hat{o} = \sum_{k=1}^{K} T_k \alpha_k o_k, \quad \text{where} \quad T_k = exp(-\sum_{i=1}^{k-1} \sigma_i \delta_i), \quad \alpha_k = 1 - exp(-\sigma_k \delta_k) \tag{2}$$

with $K$ representing the total sample points along the light ray shooting from a pixel, $\sigma_i$ representing the learned density of the $i^{th}$ sample point, $\delta_k$ representing the distance between the $(k+1)^{th}$ and $k^{th}$ sample points. From this projection formulation, we can easily obtain 2D masks of 3D object codes given the pose and camera parameters of any query viewing angles.

**Object Code Supervision:** As shown in the right part of Figure 2, having projected 2D object predictions at hand, we choose 2D images with object annotations for supervision. However, there are two issues: 1) The number and order of ground truth objects can be very different across different views due to visual occlusions. For example, as to the same 3D object in space, its object annotation in image #1 can be quite different from its annotation in image #2. Therefore, it is non-trivial to consistently utilize 2D annotations for supervision. 2) The 2D annotations only provide labels for 3D solid objects, as non-occupied 3D space is never recorded in 2D images. Therefore, it is impossible to directly supervise non-occupied space, *i.e.*, the last dimension of $\hat{o}$, from 2D annotations. These issues make adaptions of existing 2D methods ineffective, *e.g.*, Mask-RCNN (He et al., 2017) or Swin Transformer (Liu et al., 2021b), fundamentally because they do not consider the consistency between 3D and 2D, but just segment objects on individual images.

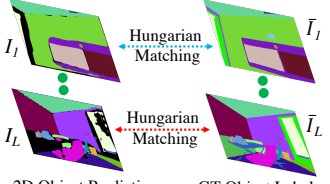

Figure 3: Illustration of 2D object matching and supervision for $\ell_{2d\_obj}$.

***To tackle the first issue***, we use the Optimal Association and Supervision strategy proposed by 3D-BoNet (Yang et al., 2019b). As illustrated in Figure 3, assuming we generate $L$ images of 2D object predictions $\{I_1 \ldots I_l \ldots I_L\}, I_l \in \mathcal{R}^{U \times V \times (H+1)}$ and have the corresponding $L$ images of 2D

ground truth object labels $\{\bar{I}_1 \ldots \bar{I}_l \ldots \bar{I}_L\}, \bar{I}_l \in \mathcal{R}^{U \times V \times T}$, in which $H$ is the predefined number of objects and $T$ represents the number of ground truth objects. Note that the ground truth number of objects in each image is usually different, but here we use the same $T$ to avoid an abuse of notation.

For each pair, we firstly take the first $H$ dimensions (solid object predictions) of $I$ and reshape them to be $M \in \mathcal{R}^{N \times H}$, where $N = U \times V$, while the last dimension is never used at this stage. Likewise, $\bar{I}$ is reshaped to be $\bar{M} \in \mathcal{R}^{N \times T}$. Then, $M$ and $\bar{M}$ are fed into Hungarian algorithm (Kuhn, 1955) to associate every ground truth 2D object mask with a unique predicted 2D object mask, according to Soft Intersection-over-Union (sIoU) and Cross-Entropy Score (CES) (Yang et al., 2019b). Formally, the Soft Intersection-over-Union (sIoU) cost between the $h^{th}$ predicted object mask and the $t^{th}$ ground truth object mask in the $l^{th}$ pair is defined as follows:

$$C_{h,t}^{sIoU} = \frac{-\sum_{n=1}^{N}(M_h^n * \bar{M}_t^n)}{\sum_{n=1}^{N} M_h^n + \sum_{n=1}^{N} \bar{M}_t^n - \sum_{n=1}^{N}(M_h^n * \bar{M}_t^n)} \quad (3)$$

where $M_h^n$ and $\bar{M}_t^n$ are the $n^{th}$ values of $M_h$ and $\bar{M}_t$. The Cross-Entropy Score (CES) between $M_h$ and $\bar{M}_t$ is formally defined as:

$$C_{h,t}^{CES} = -\frac{1}{N} \sum_{n=1}^{N} \left[ \bar{M}_t^n \log M_h^n + (1 - \bar{M}_t^n) \log(1 - M_h^n) \right] \quad (4)$$

After the optimal association based on $(C_{h,t}^{sIoU} + C_{h,t}^{CES})$, we reorder the predicted object masks to align with the $T$ ground truth masks, and then we directly minimize the cost of all ground truth objects in every pair. The final loss $\ell_{2d\_obj}$ is defined by averaging across all $L$ image pairs.

$$\ell_{2d\_obj} = \frac{1}{L} \sum_{l=1}^{L}(sIoU_l + CES_l), \text{ where } sIoU_l = \frac{1}{T} \sum_{t=1}^{T}(C_{t,t}^{sIoU}), \ CES_l = \frac{1}{T} \sum_{t=1}^{T}(C_{t,t}^{CES}) \quad (5)$$

***To tackle the second issue***, we turn to supervise the non-occupied object code in 3D space with the aid of estimated surface distances. In particular, given a specific query light ray on which we sample $K$ 3D points to compute the projected 2D object code $\hat{o}$, we simultaneously compute an approximate distance $d$ between camera center and the surface hit point along that query light:

$$d = \sum_{k=1}^{K} T_k \alpha_k \delta_k, \quad \text{where} \quad T_k = exp(-\sum_{i=1}^{k-1} \sigma_i \delta_i), \quad \alpha_k = 1 - exp(-\sigma_k \delta_k) \quad (6)$$

where $K$ represents the total sample points along light ray, $\sigma_i$ is the learned density, and $\delta_k$ is the distance between $(k+1)^{th}$ and $k^{th}$ sample points, as same as Equation 2.

As shown in Figure 4, once we have the surface distance $d$ at hand, we can easily know the relative position between every $k^{th}$ sample point and the surface point $s$ along the light ray. Naturally, we can then identify the subset of sample points surely belonging to empty space as indicated by green points, the subset of sample points near the surface as indicated by red points, and the remaining subset of sample points behind the surface as indicated by black points. Such geometric information provides critical signals to supervise empty space, *i.e.*, the last dimension of object code $o$. Note that, the sample points behind the surface may not surely belong to empty space, and therefore cannot be used as supervision signals. We use the following kernel functions to obtain a surfaceness score $s_k$ and an emptiness score $e_k$ for the $k^{th}$ sample point with the indicator function represented by $\mathbb{1}()$.

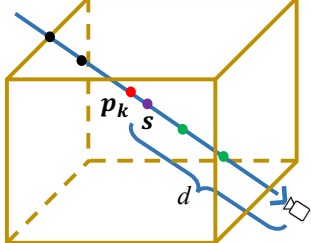

Figure 4: Empty points identification.

$$s_k = exp\left(-(d_k - d)^2\right), \quad e_k = (1 - s_k) * \mathbb{1}(d - \Delta d - d_k > 0) \quad (7)$$

where $d_k$ represents the distance between camera center and the $k^{th}$ sample point, and $\Delta d$ is a hyperparameter to compensate the inaccuracy of estimated surface distance $d$. Note that, the indicator function is defined as: $\mathbb{1}() = 1$ if $d - \Delta d - d_k > 0$, and otherwise $\mathbb{1}() = 0$. It is used to mask out the sample points behind surface point during loss calculation. Given the emptiness and surfaceness scores for the total $K$ sample points along the ray, we use the simple log loss to supervise the last dimension of object code denoted as $o_k^{H+1}$.

$$\ell_{3d\_empty} = -\frac{1}{K} \sum_{k=1}^{K} \left( e_k * log(o_k^{H+1}) + s_k * log(1 - o_k^{H+1}) \right) \quad (8)$$

Since there should be no solid object at all in the empty space, we apply the following $\ell_{3d\_obj}$ loss on the first $H$ dimensions of the object code to push them to be zeros, being complementary to the existing $\ell_{2d\_obj}$ loss. The $h^{th}$ dimension of the $k^{th}$ sample point's object code is denoted by $o_k^h$.

$$\ell_{3d\_obj} = -\frac{1}{K} \sum_{k=1}^{K} \left( e_k * \sum_{h=1}^{H} log(1 - o_k^h) \right) \qquad (9)$$

To sum up, we firstly project object codes in 3D space back to 2D pixels along light rays using volume rendering equation, and then use optimal association strategy to compute the loss value with 2D object labels only. In addition, we introduce the key emptiness and surfaceness scores for points in 3D space with the aid of estimated surface distances. These unique scores are used to supervise the non-occupied 3D space. The whole object field is jointly supervised by:

$$\ell = \ell_{2d\_obj} + \ell_{3d\_empty} + \ell_{3d\_obj} \qquad (10)$$

### 3.2 OBJECT MANIPULATOR

If we want to manipulate a specific 3D object shape (*e.g.,* translation, rotation and size adjustment), how does the whole scene look like in a new perspective after manipulation, assuming the object code and manipulation matrix are precomputed from users' interactions (*e.g.,* click, drag or zoom)?

Intuitively, there could be two strategies: 1) firstly project 3D scene into 2D images, and then edit objects in 2D space. 2) firstly edit objects in 3D space, and then project into 2D images. Compared with the first strategy which would inevitably incur inconsistency across multiple images due to the independent edits on individual views, the latter is more favourable. Remarkably, our object field component can support the latter strategy. Regarding such a manipulation task on implicit fields, the core question is: how do we edit the codes $\sigma$/$c$/$o$ of every sample point along the query light ray, such that the generated novel view exactly shows the new appearance? This is nontrivial as:

- First, we need to address potential collisions between objects during manipulation. This is quite intuitive, thanks to our special design of emptiness score in the last dimension of object code $o$.
- Second, due to visual occlusions, object codes behind surface points may not be accurate as they are not sufficiently optimized. By comparison, the projected object code $\hat{o}$ along a light ray tends to be more accurate primarily because we have ground truth 2D labels for strong supervision.
- At last, we need a systematic procedure to update the codes with the known manipulation information. To this end, we design an inverse query approach.

**Inverse Query:** We design a creative inverse query approach to address all above issues, realizing the favourable strategy: *editing in 3D space followed by 2D projection*. In particular, as shown in Figure 5, for any 3D sample point $p_k$ along a specific query light ray, given the *target* (*i.e.*, to-be-edited) object code $o_t$ and its manipulation settings: relative translation $\Delta p$=$(\Delta x, \Delta y, \Delta z)$, rotation matrix $\mathbf{R}^{3 \times 3}$, and scaling factor $t$>$0$, we firstly compute an inverse 3D point $p_{k'}$, and then evaluate whether $p_k$ and $p_{k'}$ belong to the target object, and lastly

Figure 5: Inverse points computation.

decide to whether edit the codes or not. Formally, we introduce Inverse Query Algorithm 1 to conduct a single light ray editing and rendering for object shape manipulation. Naturally, we can shoot a bunch of rays from any novel viewing angles to generate images of manipulated 3D scenes.

### 3.3 IMPLEMENTATION

To preserve the high-quality of image rendering, our loss in Equation 10 is only used to optimize the MLPs of object field branch. The backbone is only optimized by the original NeRF photo-metric loss (Mildenhall et al., 2020). The whole network is end-to-end trained from scratch. The single hyper-parameter for our object field $\Delta d$ is set as 0.05 meters in all experiments.

## 4 EXPERIMENTS

### 4.1 DATASETS

**DM-SR:** To the best of our knowledge, there is no 3D scene dataset suitable for quantitative evaluation of geometry manipulation. Therefore, we create a **s**ynthetic dataset with 8 different and complex indoor **r**ooms, called DM-SR. For each scene, we generate the following 5 groups of images:

- Group 1 (w/o Manipulation): Color images and 2D object masks at $400 \times 400$ pixels are rendered from viewpoints on the upper hemisphere. We generate 300 views for training.

---

**Algorithm 1** Our Inverse Query Algorithm to manipulate the learned implicit fields. (1) $\boldsymbol{o}_t$ is the target object code, one-hot with $H+1$ dimensions. $\{\Delta \boldsymbol{p}, \mathbf{R}^{3\times 3}, t>0\}$ represent the manipulation information for the target object. (2) $\{\boldsymbol{p}_1 \cdots \boldsymbol{p}_k \cdots \boldsymbol{p}_K\}$ represent the $K$ sample points along a specific query light ray $\boldsymbol{r}$. Note that, we convert all object codes into hard one-hot vectors for easy implementation.

---

**Input:**
- The target object code $\boldsymbol{o}_t$, manipulation information $\{\Delta \boldsymbol{p}, \mathbf{R}^{3\times 3}, t \geq 0\}$;
- The sample points $\{\boldsymbol{p}_1 \cdots \boldsymbol{p}_k \cdots \boldsymbol{p}_K\}$ along a specific query light ray $\boldsymbol{r}$;

**Output:**
- The final pixel color $\bar{c}$ rendered from the query ray $\boldsymbol{r}$ after manipulation;
- The final pixel object code $\bar{\boldsymbol{o}}$ rendered from the query ray $\boldsymbol{r}$ after manipulation;

**Preliminary step:**
- Obtain the projected pixel object code $\hat{\boldsymbol{o}}$ of light ray $\boldsymbol{r}$ before manipulation;

*NOTE: The loop below shows how to edit per sample point in 3D space.*

**for** $\boldsymbol{p}_k$ in $\{\boldsymbol{p}_1 \cdots \boldsymbol{p}_k \cdots \boldsymbol{p}_K\}$ **do**
- Compute the inverse point $\boldsymbol{p}_{k'}$ for $\boldsymbol{p}_k$:    $\boldsymbol{p}_{k'} = (1/t)\mathbf{R}^{-1}(\boldsymbol{p}_k - \Delta \boldsymbol{p})$;
- Obtain the codes $\{\sigma_k, \boldsymbol{c}_k, \boldsymbol{o}_k\}$ for the point $\boldsymbol{p}_k$;
- Obtain the codes $\{\sigma_{k'}, \boldsymbol{c}_{k'}, \boldsymbol{o}_{k'}\}$ for the inverse point $\boldsymbol{p}_{k'}$;
- *Tackle visual occlusions:*
    if $\boldsymbol{o}_k = \boldsymbol{o}_t$ and $\boldsymbol{o}_k \neq \hat{\boldsymbol{o}}$ do:    $\boldsymbol{o}_k \leftarrow \hat{\boldsymbol{o}}$
    *Note: the target object is behind the surface but will be manipulated;*
- Obtain new implicit codes $\{\bar{\sigma}_k, \bar{\boldsymbol{c}}_k, \bar{\boldsymbol{o}}_k\}$ for $\boldsymbol{p}_k$ after manipulation:
    if $\boldsymbol{o}_k \neq \boldsymbol{o}_t$ and $o_k^{H+1} \neq 1$ and $\boldsymbol{o}_{k'} = \boldsymbol{o}_t$ do:    collision detected, EXIT.
    if $\boldsymbol{o}_k \neq \boldsymbol{o}_t$ and $\boldsymbol{o}_{k'} = \boldsymbol{o}_t$ do:    $\{\bar{\sigma}_k, \bar{\boldsymbol{c}}_k, \bar{\boldsymbol{o}}_k\} \leftarrow \{\sigma_{k'}, \boldsymbol{c}_{k'}, \boldsymbol{o}_{k'}\}$ ;
    if $\boldsymbol{o}_k = \boldsymbol{o}_t$ and $\boldsymbol{o}_{k'} = \boldsymbol{o}_t$ do:    $\{\bar{\sigma}_k, \bar{\boldsymbol{c}}_k, \bar{\boldsymbol{o}}_k\} \leftarrow \{\sigma_{k'}, \boldsymbol{c}_{k'}, \boldsymbol{o}_{k'}\}$ ;
    if $\boldsymbol{o}_k = \boldsymbol{o}_t$ and $\boldsymbol{o}_{k'} \neq \boldsymbol{o}_t$ do:    $\{\bar{\sigma}_k, \bar{\boldsymbol{c}}_k, \bar{\boldsymbol{o}}_k\} \leftarrow \{0, \boldsymbol{0}, \boldsymbol{0}\}$ ;
    if $\boldsymbol{o}_k \neq \boldsymbol{o}_t$ and $\boldsymbol{o}_{k'} \neq \boldsymbol{o}_t$ do:    $\{\bar{\sigma}_k, \bar{\boldsymbol{c}}_k, \bar{\boldsymbol{o}}_k\} \leftarrow \{\sigma_k, \boldsymbol{c}_k, \boldsymbol{o}_k\}$ ;

After the above *for loop*, every point $\boldsymbol{p}_k$ will get new implicit codes $\{\bar{\sigma}_k, \bar{\boldsymbol{c}}_k, \bar{\boldsymbol{o}}_k\}$.
*NOTE: The step below shows how to project edited 3D points to a 2D image.*
According to volume rendering equation, the final pixel color and object code are:
- $\bar{c} = \sum_{k=1}^{K} \bar{T}_k \bar{\alpha}_k \bar{\boldsymbol{c}}_k, \quad \bar{\boldsymbol{o}} = \sum_{k=1}^{K} \bar{T}_k \bar{\alpha}_k \bar{\boldsymbol{o}}_k$
    *where* $\bar{T}_k = exp(-\sum_{i=1}^{k-1} \bar{\sigma}_i \delta_i), \quad \bar{\alpha}_k = 1 - exp(-\bar{\sigma}_k \delta_k)$.

---

- Group 2 (Translation Only): One object is selected to be translated along $x$ or $y$ axis with $\sim 0.3m$.
- Group 3 (Rotation Only): One object is selected to be rotated around $z$ axis with about 90 degrees.
- Group 4 (Scaling Only): One object is selected to be scaled down about $0.8\times$ smaller.
- Group 5 (Joint Translation/Rotation/Scaling): One object is selected to be simultaneously translated about $\sim 0.3m$, rotated about 90 degrees, scaled down about $0.8\times$ smaller.

For each group, 100 views are generated for testing at the same viewpoints.

**Replica:** Replica (Straub et al., 2019) is a reconstruction-based 3D dataset of high fidelity scenes. We request the authors of Semantic-NeRF (Zhi et al., 2021a) to generate (180 training / 180 testing) color images and 2D object masks with camera poses at 640×480 pixels for each of 7 scenes. Each scene has 59~93 objects with very diverse sizes.

**ScanNet:** ScanNet (Dai et al., 2017) is a large-scale challenging real-world dataset. We select 8 scenes (~10 objects in each one) for evaluation. Each scene has ~3000 raw images with 2D object masks and camera poses, among which we evenly select 300 views for training and 100 for testing.

## 4.2 BASELINE AND METRICS

**Scene Decomposition:** The most relevant work to us is ObjectNeRF (Yang et al., 2021), but its design is vastly different: 1) It requires a point cloud as input for voxelization in training, but we do not. 2) It needs GT bounding boxes of target objects to manually prune point samples during editing, but we do not need any annotations in editing. 3) It only learns to binarily segment the foreground object and background, by pre-defining an Object Library in training and editing. However, our object code is completely learned from scratch. This means that ObjectNeRF is not comparable due to the fundamental differences. Note that, the recent Semantic-NeRF (Zhi et al., 2021a) is also not comparable because it only learns 3D semantic categories, not individual 3D objects. In fact, we notice that all recent published relevant works (Yang et al., 2021; Zhi et al., 2021a; Wu et al., 2022; Kundu et al., 2022; Norman et al., 2022; Fu et al., 2022) do not directly tackle and evaluate multiple 3D object segmentation in literature. In this regard, we use the powerful Mask-RCNN (He et al., 2017) and Swin Transformer (Liu et al., 2021b) as baselines.

Since we train our DM-NeRF in scene-specific fashion, for fairness, we also fine-tune Mask-RCNN and Swin Transformer models (pretrained under Detectron2 Library) on every single scene. In particular, we carefully fine-tune both models using up to 480 epochs until convergence with learning rate $5e^{-4}$ and then pick up the best models on the testing split of each scene for comparison.

**Object Manipulation:** Since there is no method that can directly manipulate objects in continuous radiance fields, we adapt the recent Point-NeRF (Xu et al., 2022) for comparison, because it can recover both explicit 3D scene point clouds and novel views. Adaptation details are in Appendix.

**Metrics:** We use the standard PSNR/SSIM/LPIPS scores to evaluate color image synthesis (Mildenhall et al., 2020), and use AP of all 2D test images to evaluate 3D scene decomposition.

### 4.3 3D Scene Decomposition

**Training with 100% Accurate 2D Labels**: We evaluate the performance of scene decomposition on 3 datasets. For DM-SR dataset, we evaluate our method and baselines on images of Group 1 only, while the images of Groups 2/3/4/5 are used for object manipulation. For every single scene in these datasets, using 100% accurate 2D object ground truth labels, we train a separate model for our method, and fine-tune separate models for Mask-RCNN (MR) and Swin Transformer (SwinT).

**Analysis**: Table 1 shows that our method, not surprisingly, achieves excellent results for novel view rendering thanks to the original NeRF backbone. Notably, our method obtains nearly perfect object segmentation results across multiple viewing points of complex 3D scenes in all three datasets, clearly outperforming baselines. Figure 10 shows that our results have much sharper object boundaries thanks to the explicit 3D geometry applied in our object field.

| | DM-SR Dataset | | | | | | Replica Dataset | | | | | | | ScanNet Dataset | | | | | |
|---|---|---|---|---|---|---|---|---|---|---|---|---|---|---|---|---|---|---|---|---|
| | Novel View Synthesis | | | Decomposition | | | | Novel View Synthesis | | | Decomposition | | | | Novel View Synthesis | | | Decomposition | | |
| | PSNR↑ | SSIM↑ | LPIPS↓ | MR | SwinT | **Ours** | | PSNR↑ | SSIM↑ | LPIPS↓ | MR | SwinT | **Ours** | | PSNR↑ | SSIM↑ | LPIPS↓ | MR | SwinT | **Ours** |
| Bathroom | 44.05 | 0.994 | 0.009 | 93.81 | 98.89 | 100.0 | | | | | | | | 0010.00 | 26.82 | 0.809 | 0.381 | 83.90 | 87.59 | 94.82 |
| Bedroom | 48.07 | 0.996 | 0.009 | 97.92 | 98.85 | 100.0 | Office_0 | 40.66 | 0.972 | 0.070 | 74.05 | 80.17 | 82.71 | 0012.00 | 29.28 | 0.753 | 0.389 | 86.90 | 89.92 | 98.86 |
| Dinning | 42.34 | 0.984 | 0.028 | 98.85 | 97.81 | 99.66 | Office_2 | 36.98 | 0.964 | 0.115 | 73.41 | 75.39 | 81.12 | 0024.00 | 23.68 | 0.705 | 0.452 | 69.87 | 67.88 | 93.25 |
| Kitchen | 46.06 | 0.994 | 0.014 | 92.04 | 98.81 | 100.0 | Office_3 | 35.34 | 0.955 | 0.078 | 72.91 | 73.26 | 76.30 | 0033.00 | 27.76 | 0.856 | 0.342 | 88.70 | 94.23 | 97.02 |
| Reception | 42.59 | 0.993 | 0.008 | 98.81 | 95.75 | 100.0 | Office_4 | 32.95 | 0.921 | 0.172 | 74.76 | 72.51 | 70.33 | 0038.00 | 29.36 | 0.716 | 0.415 | 96.01 | 97.94 | 99.17 |
| Rest | 42.80 | 0.994 | 0.007 | 98.89 | 94.50 | 99.89 | Room_0 | 34.97 | 0.940 | 0.127 | 78.67 | 76.90 | 79.83 | 0088.00 | 29.37 | 0.825 | 0.386 | 69.06 | 81.63 | 83.59 |
| Study | 41.08 | 0.987 | 0.026 | 96.86 | 97.88 | 99.86 | Room_1 | 34.72 | 0.931 | 0.134 | 78.38 | 81.41 | 92.11 | 0113.00 | 31.19 | 0.878 | 0.320 | 98.59 | 98.12 | 98.67 |
| Office | 46.38 | 0.996 | 0.006 | 97.83 | 96.87 | 100.0 | Room_2 | 37.32 | 0.963 | 0.115 | 77.58 | 80.33 | 84.78 | 0192.00 | 28.19 | 0.732 | 0.376 | 96.95 | 98.24 | 99.40 |
| Average | 44.17 | 0.992 | 0.013 | 96.87 | 97.42 | **99.80** | Average | 36.13 | 0.949 | 0.116 | 75.68 | 77.14 | **81.03** | Average | 28.21 | 0.784 | 0.383 | 86.25 | 89.71 | **95.60** |

Table 1: Quantitative results on three datasets. The metric for object decomposition is $AP^{0.75}$.

**Robustness to Noisy 2D Labels**: Since our method inherently has multi-view consistency while the 2D segmentation methods do not, it is expected that our method has better robustness to inaccurate and noisy 2D labels in training. To validate this advantage, we conduct the following experiments on DM-SR dataset. Particularly, as illustrated in Figure 6, we randomly assign incorrect object labels to different amounts of image pixels of all training images (10%/50%/70%/80%/90%), and then our method and baselines are all trained with these noisy 2D labels.

**Analysis**: As shown in Table 2, it can be seen that our method still achieves an excellent object segmentation score ($AP^{0.75}$ = 74.08) on testing/novel views, even though 80% of 2D labels are incorrect in training. By contrast, both baselines fail catastrophically once more than 50% of labels are noisy in training. Basically, this is because our dedicated losses $\ell_{2d\_obj}/\ell_{3d\_empty}/\ell_{3d\_obj}$ explicitly take into account object geometry consistency across multi-views, thus allowing the estimated 3D object codes to be resistant against wrong labels, whereas the existing 2D object segmentation methods only independently process single images, being easily misled by wrong labels in training. This clearly demonstrates the remarkable robustness of our method. More experiments and quantitative/qualitative results regarding robustness are in Appendix.

**Extension to Panoptic Segmentation:** Our method can be easily extended to tackling panoptic segmentation by adding an extra semantic branch parallel to object code branch. Due to the limited space, quantitative and qualitative results are provided in Appendix.

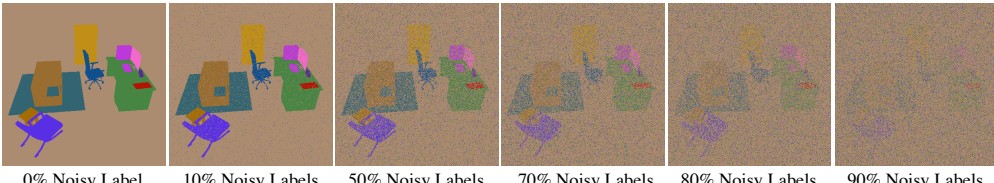

| 0% Noisy Label | 10% Noisy Labels | 50% Noisy Labels | 70% Noisy Labels | 80% Noisy Labels | 90% Noisy Labels |

Figure 6: Examples of 2D object labels with different levels of noise.

| | 10% Noisy Labels | | | 50% Noisy Labels | | | 70% Noisy Labels | | | 80% Noisy Labels | | | 90% Noisy Labels | | |
|---|---|---|---|---|---|---|---|---|---|---|---|---|---|---|---|
| | MR | SwinT | **Ours** | MR | SwinT | **Ours** | MR | SwinT | **Ours** | MR | SwinT | **Ours** | MR | SwinT | **Ours** |
| Bathroom | 98.93 | 98.96 | 99.81 | 54.52 | 62.94 | 99.63 | 1.86 | 7.71 | 99.02 | 1.89 | 7.78 | 58.09 | 1.79 | 7.11 | 9.69 |
| Bedroom | 98.61 | 98.85 | 100.0 | 75.41 | 92.94 | 100.0 | 3.00 | 3.96 | 100.0 | 2.96 | 3.18 | 82.83 | 2.95 | 2.75 | 4.25 |
| Dinning | 95.57 | 97.88 | 98.41 | 46.52 | 43.50 | 85.48 | 1.40 | 1.05 | 81.91 | 1.51 | 1.09 | 63.50 | 1.36 | 0.96 | 14.44 |
| Kitchen | 98.82 | 98.81 | 100.0 | 84.93 | 93.60 | 100.0 | 4.09 | 8.42 | 100.0 | 4.02 | 5.04 | 51.87 | 4.04 | 4.69 | 1.80 |
| Reception | 80.96 | 91.00 | 100.0 | 32.78 | 42.11 | 100.0 | 1.03 | 6.27 | 100.0 | 0.73 | 1.96 | 100.0 | 0.22 | 1.64 | 37.63 |
| Rest | 93.58 | 94.50 | 99.64 | 52.75 | 51.99 | 99.32 | 1.63 | 1.57 | 99.33 | 2.46 | 2.02 | 66.74 | 1.63 | 1.69 | 11.11 |
| Study | 93.07 | 97.94 | 98.58 | 49.03 | 60.04 | 97.97 | 1.44 | 6.24 | 98.03 | 1.50 | 6.06 | 72.62 | 1.40 | 5.48 | 40.72 |
| Office | 95.09 | 97.00 | 100.0 | 66.28 | 69.11 | 100.0 | 4.13 | 2.96 | 100.0 | 2.61 | 3.94 | 97.00 | 2.66 | 2.48 | 0.00 |
| Average | 94.33 | 96.87 | **99.56** | 57.78 | 64.53 | **97.80** | 2.32 | 4.77 | **97.29** | 2.21 | 3.88 | **74.08** | 2.00 | 3.35 | **14.96** |

Table 2: Quantitative object decomposition $AP^{0.75}$ scores on noisy DM-SR dataset.

| | Translation | | | | Rotation | | | |
|---|---|---|---|---|---|---|---|---|
| | PSNR↑ | SSIM↑ | LPIPS↓ | $AP^{0.9}$ ↑ | PSNR↑ | SSIM↑ | LPIPS↓ | $AP^{0.9}$ ↑ |
| Point-NeRF | 25.79 | 0.847 | 0.202 | - | 25.21 | 0.818 | 0.203 | - |
| Ours | **33.94** | **0.975** | **0.033** | **89.33** | **31.94** | **0.969** | **0.038** | **85.68** |
| Ablation 1: Ours (w/o $\ell_{3d}$) | 32.84 | 0.967 | 0.048 | 87.26 | 30.38 | 0.945 | 0.090 | 82.46 |
| Ablation 2: Ours (w/o VO) | 33.54 | 0.970 | 0.045 | 86.93 | 30.57 | 0.953 | 0.076 | 82.43 |
| | Scale | | | | Joint | | | |
| | PSNR↑ | SSIM↑ | LPIPS↓ | $AP^{0.9}$ ↑ | PSNR↑ | SSIM↑ | LPIPS↓ | $AP^{0.9}$ ↑ |
| Point-NeRF | 25.83 | 0.848 | 0.202 | - | 23.51 | 0.816 | 0.226 | - |
| Ours | **33.40** | **0.971** | **0.037** | **86.05** | **30.65** | **0.965** | **0.045** | **81.70** |
| Ablation 1: Ours (w/o $\ell_{3d}$) | 31.84 | 0.959 | 0.062 | 83.31 | 29.95 | 0.947 | 0.088 | 77.36 |
| Ablation 2: Ours (w/o VO) | 32.43 | 0.964 | 0.054 | 76.33 | 29.85 | 0.951 | 0.075 | 74.71 |

Table 3: Quantitative results of object manipulation and ablation studies on DM-SR dataset.

## 4.4 3D OBJECT MANIPULATION

In this section, we directly use our model trained on the images of Group 1 to test on the remaining images of Groups 2/3/4/5 in DM-SR dataset. In particular, with the trained model, we feed the known manipulation information of Groups 2/3/4/5 into Algorithm 1, generating images and 2D object masks. These (edited) images and masks are compared with the ground truth 2D views. For comparison, we carefully train Point-NeRF models in scene-specific fashion and apply the same manipulation information to its learned point clouds followed by novel view rendering.

**Analysis**: Table 3 shows that the quality of our novel view rendering is clearly better than Point-NeRF (Xu et al., 2022), although it decreases after manipulation compared with non-manipulation in Table 1, primarily because the lighting factors are not decomposed and the illumination of edited objects shows discrepancies. However, the object decomposition is still nearly perfect, as also shown for deformation manipulation in Figure 11. More results are in Appendix.

## 4.5 ABLATION STUDY

To evaluate the effectiveness of our key designs of object field and manipulator, we conduct two ablation studies. 1) We only remove the loss functions $(\ell_{3d\_empty} + \ell_{3d\_obj})$ which are jointly designed to learn correct codes for empty 3D points, denoted as w/o $\ell_{3d}$. 2) During manipulation, we remove the step "Tackle visual occlusions" in Algorithm 1, denoted as w/o VO. As shown in Table 3, both of our loss functions for empty space regularization and visual occlusion handling step are crucial for accurate 3D scene decomposition and manipulation. More ablation results are in Appendix.

## 5 DISCUSSION AND CONCLUSION

We have shown that it is feasible to simultaneously reconstruct, decompose, manipulate and render complex 3D scenes in a single pipeline only from 2D views. By adding an object field component into the implicit representation, we successfully decompose all individual objects in 3D space. The decomposed object shapes can be further freely edited by our visual occlusion aware manipulator.

One limitation is the lack of decomposing lighting factors, which is non-trivial and left for future work. In addition, manipulation of originally occluded objects or parts may produce artifacts due to the inaccuracy of learned radiance fields, although these artifacts can be easily repaired by applying simple heuristics such as continuity of surfaces or using better neural backbones.

**Acknowledgements:** This work was supported in part by National Natural Science Foundation of China under Grant 62271431, in part by Shenzhen Science and Technology Innovation Commission under Grant JCYJ20210324120603011, in part by Research Grants Council of Hong Kong under Grants 25207822 & 15225522.

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
