# OpenReview forum: "DM-NeRF: 3D Scene Geometry Decomposition and Manipulation from 2D Images"
_ICLR.cc/2023/Conference — ICLR 2023 poster_

### Official Review · Reviewer_Tr2a · 2022-10-24

**Confidence:** 4
**Correctness:** 2
**Technical Novelty And Significance:** 2
**Empirical Novelty And Significance:** 2
**Recommendation:** 6

**Clarity, Quality, Novelty And Reproducibility:**

See strength and weaknesses. I don't think it is easily reproducible given the unclear notation and loss functions (mentioned in minors).

**Strength And Weaknesses:**

Strength
- The problem of 3D object segmentation from images is an important problem, and this paper takes an interesting step to solve it.
- Novelty. The Hungarian matching-based loss seems to be an interesting solution to handle the 2D instance label mismatch during loss evaluation. However, the paper did not provide much detail. For instance, what is the cost function when matching the rendered labels to the ground-truth. When it may fail?

Weakness
- Clarity. My major concern is on writing and clarity. The method is hard to keep track of, and I don't think readers can easily understand how it work. To improve the writing, the authors may want to improve the figures, use precise notation for losses, and put adequate emphasis on certain parts. See minor comments for more.
- Experiments. I found the comparison against over-fitted mask-rcnn to be not informative. Assuming the ground-truth segmentation is perfect, mask-rcnn fine-tuned per scene could easily overfit to the ground-truth labels and produce high accuracy. The fact it did not produce high accuracy made me think it is not well-trained. I can only see this experiment to be meaningful if the input segmentation is noisy, and the paper aims to clean it up, while mask-rcnn is not able to do so.
- Generalization to real data and other setups. The method seemed to be trained on synthetic data with good instance segmentation. If we were to apply it to real world images, we need to answer how sensitive is the method to bad 2D instance segmentations. A related question is how to handle results of existing detectors (for instance Mask-RCNN only gives instance segmentation but not the background and stuff classes). Does it work with panoptic segmentation?

Minor comments
- Fig 1: object manipulation block is left empty, is this intended?
- Fig 2: losses appeared but not discussed or referred to. Ideally, by looking at the figure, readers would be able to understand the gist of the method.
- Fig 3: does this corresponds to l-2d-obj? Please make it clear in the caption.
- Eq 4: $\delta_k$ is not defined. Is it the same as $d_k$? If $\delta_k$ is the depth, Eq 4 shouldn’t be called surface distance since it’s measuring the distance to the closest surface along the ray instead of the closest surface in the scene.
- Eq 5: how does the indicator function work here? If $d-\Delta d-d_k =0$, value is one?
- Eq 7: The name of $l_{3d-obj}$ is slightly confusing as it optimizes the emptiness of the object label, but the name indicates it has similar functionality as $l_{2d-obj}$, which optimizes the object label.

**Summary Of The Paper:**

The paper studies the problem of 3D object segmentation and manipulation from images with instance segmentations. It uses NeRF and ask it to further produce an object label per 3D point. They introduce a Hungarian matching based loss to supervise the object label fields with 2D instance segmentations (although that part wasn't clear in the paper). They show better results than the 2D mask-rcnn baseline in terms of 2d segmentation.

**Summary Of The Review:**

Overall, the problem setup is novel and the solution is new. However, the method is not clearly described and the comparison to baseline is not well-designed.

---

> ### Author Response · Authors · 2022-11-20
> **Response to Reviewer Tr2a**
>
> **Q2: Clarity. My major concern is on writing and clarity. The method is hard to keep track of, and I don't think readers can easily understand how it work. To improve the writing, the authors may want to improve the figures, use precise notation for losses, and put adequate emphasis on certain parts. See minor comments for more.**
>
> **A2:** We really appreciate the reviewer's time to identify these issues, which are all clearly fixed in the revised paper. In particular, Figs 1/2/3 are updated with better captions, and Eqs 6/7/9 (previously 4/5/7) are updated with clear explanations.
>
> **Q3: Experiments. I found the comparison against over-fitted mask-rcnn to be not informative. Assuming the ground-truth segmentation is perfect, mask-rcnn fine-tuned per scene could easily overfit to the ground-truth labels and produce high accuracy. The fact it did not produce high accuracy made me think it is not well-trained. I can only see this experiment to be meaningful if the input segmentation is noisy, and the paper aims to clean it up, while mask-rcnn is not able to do so.**
>
> **A3:** Thank you for the great advice. In the revised paper, in Section 4.3, we conduct extra extensive experiments on 5 variants of noisy DM-SR dataset to evaluate the robustness of our method regarding noisy 2D object labels during training. Tables 1\&2 present quantitative results.
>
> We can see that our method is clearly more accurate and robust than both Swin Transformer and MaskRCNN for 3D scene decomposition. Figure 9 of the Appendix shows qualitative results of our models trained with different amounts of noisy 2D labels. Notably, given 80\% of 2D object labels with noise on our DM-SR dataset, both Swin Transformer and MaskRCNN fail catastrophically (AP$^{0.75}<$4\%), while our method still achieves AP$^{0.75}$ = 74.08\% in object segmentation results.
>
> Overall, as the reviewer expected, our method is indeed able to give clean object decomposition results, while MaskRCNN and Swin Transformer cannot. This is because the 2D baselines only take into account separate images in training, while our method inherently has multi-view consistency which allows the model to be resistant to noisy labels.
>
> **Q4: Generalization to real data and other setups. The method seemed to be trained on synthetic data with good instance segmentation. If we were to apply it to real world images, we need to answer how sensitive is the method to bad 2D instance segmentations. A related question is how to handle results of existing detectors (for instance Mask-RCNN only gives instance segmentation but not the background and stuff classes).**
>
> **A4:** This is a great point. As requested, we train our method directly on inaccurate 2D labels (estimated by MaskRCNN). As shown in Figure 10 of Appendix, our method still achieves excellent object decomposition results, except that the thin chair feet are not segmented. This result is consistent with that of noisy 2D labels in Section 4.3, showing the remarkable robustness of our method and the applicability in real world. Due to the limited time, more complete experimental results will be out within one week.
>
> **Q5: Does it work with panoptic segmentation?**
>
> **A5:** The answer is YES. As requested, an extra semantic segmentation branch parallel to object code branch is added into our current DM-NeRF. As shown in Figure 13 of Appendix, it can be seen that both semantic categories and object codes are accurately inferred. Due to the limited time, more complete experimental results will be out within one week.
>
> **Q6: Minor comments.**
>
> **A6:** We really appreciate the reviewer's time to identify these issues, which are all clearly fixed in the revised paper. In particular, Figs 1/2/3 are updated with better captions, and Eqs 6/7/9 (previously 4/5/7) are updated with clear explanations.
>
> **Q7: See strength and weaknesses. I don't think it is easily reproducible given the unclear notation and loss functions (mentioned in minors).**
>
> **A7:** The main paper is carefully revised and significantly improved according to all reviewers' suggestions.  For reproducing all results and benefiting the community, our full code will be made public and is currently anonymous at [https://github.com/DM-NeRF-ICLR-Rebuttal/DM-NeRF](https://github.com/DM-NeRF-ICLR-Rebuttal/DM-NeRF).
>
> **Q8: Overall, the problem setup is novel and the solution is new. However, the method is not clearly described and the comparison to baseline is not well-designed.**
>
> **A8:** Thank you for appreciating the novelties of our paper. In the revised paper, our method especially the Hungarian matching loss is clearly described, and we add two more baselines for comparing 3D scene decomposition and object manipulation. We additionally evaluate the robustness of our method over noisy and inaccurate 2D training labels, and the feasibility of panoptic segmentation.
>
> Above all, we believe that our current version is ready and deliverable.

---

> > ### Author Response · Authors · 2022-11-28
> > **Additional Quantitative Results**
> >
> > To further support our previous reponses, additional quantitative results are presented as follows:
> >
> > **1. Inaccurate Supervision Signals (Q4)**
> >
> > We use the instance masks estimated by Mask-RCNN as the supervision signals when training our DM-NeRF. From Table 5, we can see that even though the 2D labels are inaccurate for training, our method still achieves excellent object decomposition results.
> >
> > **Table 5: Quantitative object decomposition AP$^{0.75}$ scores of our method trained on inaccurate 2D labels (estimated by MaskRCNN).**
> > |         |          |         |         |         |           |       |       |        |         |
> > |---------|----------|---------|---------|---------|-----------|-------|-------|--------|---------|
> > | Scenes  | Bathroom | Bedroom | Dinning | Kitchen | Reception | Rest  | Study | Office | Average |
> > | Ours_MR | 95.30    | 97.08   | 95.69   | 94.72   | 99.62     | 98.58 | 97.72 | 98.08  | 97.10   |
> > |         |          |         |         |         |           |       |       |        |         |
> >
> > **2. Panoptic Segmentation (Q5)**
> >
> > An extra semantic branch parallel to object code branch is added into our current DM-NeRF for panoptic segmentation. From Table 6, it can be seen that both semantic categories [Sem (IoU↑)] and object codes [Obj (AP$^{0.75}$↑)]  are accurately inferred.
> >
> > **Table 6: Quantitative results of panoptic segmentation on DM-SR dataset.**
> > |           |          |          |
> > |----------:|:--------:|:--------:|
> > |           |  Obj (AP$^{0.75}$↑)   |   Sem (IoU↑)  |
> > |  Bathroom |   100.0  |   97.58  |
> > |   Bedroom |   100.0  |   99.08  |
> > |   Dinning |   99.41  |   94.64  |
> > |   Kitchen |   100.0  |   98.72  |
> > | Reception |   100.0  |   97.42  |
> > |      Rest |   97.86  |   97.13  |
> > |     Study |   96.84  |   94.29  |
> > |    Office |   100.0  |   97.85  |
> > |   Average |   99.26  |   97.09  |
> > |           |          |          |

---

> > > ### Comment · Reviewer_Tr2a · 2022-12-13
> > > **Thanks for the rebuttal. My rating is updated to 6.**
> > >
> > > Thanks for the rebuttal. It addressed most of my concerns.
> > > - The results and comparisons given noisy input 2D segmentation are strong.
> > > - The writing is clearer.

---

> > > > ### Author Response · Authors · 2022-12-14
> > > > **Thanks**
> > > >
> > > > Dear reviewer Tr2a,
> > > >
> > > > Thank you for your positive and encouraging feedback. Your valuable comments and suggestions truly helped us significantly improve our manuscipt.
> > > >
> > > > Regards,
> > > > Authors

---

> ### Author Response · Authors · 2022-11-20
> **Response to Reviewer Tr2a**
>
> We appreciate the reviewer's insightful comments and address all concerns below. Would you also check out our revised main paper (highlighted in yellow), and supplied materials (new appendix + new video demo + code) anonymous at [https://github.com/DM-NeRF-ICLR-Rebuttal/DM-NeRF](https://github.com/DM-NeRF-ICLR-Rebuttal/DM-NeRF).
>
> **Q1: Novelty. The Hungarian matching-based loss seems to be an interesting solution to handle the 2D instance label mismatch during loss evaluation. However, the paper did not provide much detail. For instance, what is the cost function when matching the rendered labels to the ground-truth. When it may fail?**
>
> **A1:** Thank you for the advice. In the revised paper, we move the details of Hungarian matching-based loss to the main paper (pages 4$\sim$5), which was provided in the original appendix. In particular, the details of cost functions are provided in Equations 3/4/5.
>
> As shown in Table 2, given 2D labels with a significant amount (90\%) of noise in training, our algorithm finally fails (AP$^{0.75}$=14.96\%). Otherwise, our optimal matching based loss actually is remarkably successful and robust, as shown in Tables 1/2.
>
> **Table 1: Quantitative object decomposition AP$^{0.75}$ scores on DM-SR dataset.**
> |||||
> |-:|:-:|:-:|:-:|
> ||   MR  | SiwnT | **Ours** |
> |  Bathroom | 93.81 | 98.89 |   100.0  |
> |   Bedroom | 97.92 | 98.85 |   100.0  |
> |   Dinning | 98.85 | 97.81 |   99.66  |
> |   Kitchen | 92.06 | 98.81 |   100.0  |
> | Reception | 98.81 | 95.74 |   100.0  |
> |      Rest | 98.89 | 94.50 |   99.89  |
> |     Study | 96.86 | 97.88 |   98.86  |
> |    Office | 97.83 | 96.87 |   100.0  |
> |   Average | 96.87 | 97.42 |   **99.80**  |
> |||||
>
> **Table 2: Quantitative object decomposition AP$^{0.75}$ scores on noisy DM-SR dataset.**
> |||||||||||||||||
> |-:|:-:|:-:|:-:|:-:|:-:|:-:|:-:|:-:|:-|:-:|:-:|:-:|:-:|:-:|:-:|
> | **Noisy Labels** | **10%** |       |           | **50%** |       |           | **70%** |       |           | **80%** |       |           | **90%** |       |           |
> ||    MR   | SiwnT |  **Ours** |    MR   | SiwnT |  **Ours** |    MR   | SiwnT |  **Ours** |    MR   | SiwnT |  **Ours** |    MR   | SiwnT |  **Ours** |
> |         Bathroom |  98.93  | 98.96 |   99.81   |  54.52  | 62.94 |   99.63   |   1.86  |  7.71 |   99.02   |   1.89  |  7.78 |   58.09   |   1.79  |  7.11 |    9.69   |
> |          Bedroom |  98.61  | 98.85 |   100.0   |  75.41  | 92.94 |   100.0   |   3.00  |  3.96 |   100.00  |   2.96  |  3.18 |   82.83   |   2.95  |  2.75 |    4.25   |
> |          Dinning |  95.57  | 97.88 |   98.41   |  46.52  | 43.50 |   85.48   |   1.40  |  1.05 |   81.91   |   1.51  |  1.09 |   63.50   |   1.36  |  0.96 |   14.44   |
> |          Kitchen |  98.82  | 98.81 |   100.0   |  84.93  | 93.60 |   100.0   |   4.09  |  8.42 |   100.0   |   4.02  |  5.04 |   51.87   |   4.04  |  4.69 |    1.80   |
> |        Reception |  80.96  | 91.00 |   100.0   |  32.78  | 42.11 |   100.0   |   1.03  |  6.27 |   100.0   |   0.73  |  1.96 |   100.0   |   0.22  |  1.64 |   37.63   |
> |     Rest |  93.58  | 94.50 |   99.64   |  52.75  | 51.99 |   99.32   |   1.63  |  1.57 |   99.33   |   2.46  |  2.02 |   66.74   |   1.63  |  1.69 |   11.11   |
> |    Study |  93.07  | 97.94 |   98.58   |  49.03  | 60.04 |   97.97   |   1.44  |  6.24 |   98.03   |   1.50  |  6.06 |   72.62   |   1.40  |  5.48 |   40.72   |
> |    Office |  95.09  | 97.00 |   100.0   |  66.28  | 69.11 |   100.0   |   4.13  |  2.96 |   100.0   |   2.61  |  3.94 |   97.00   |   2.66  |  2.48 |    0.00   |
> |  Average |  94.33  | 96.87 | **99.56** |  57.78  | 64.53 | **97.80** |   2.32  |  4.77 | **97.29** |   2.21  |  3.88 | **74.08** |   2.00  |  3.35 | **14.96** |
> |||||||||||||||||

---

### Official Review · Reviewer_1RkG · 2022-10-25

**Confidence:** 5
**Correctness:** 3
**Technical Novelty And Significance:** 3
**Empirical Novelty And Significance:** 3
**Recommendation:** 3

**Clarity, Quality, Novelty And Reproducibility:**

Most of it is on point, but there is one missing link between the 3 layers of contributions, as stated above, which currently makes it stlightly of a tough sell without this missing justification or experimental addition.

**Details Of Ethics Concerns:**

None in particular.

**Strength And Weaknesses:**

** Strengths:

- (1) Readability.

In its current state, the paper is very well written. The main key ideas are well explained, motivated, justified and articulated.

- (2) The relative positioning is very clearly and explicitely established.

This includes multiple discussions, in particular the differential wrt other existing object-centric NERF prior work.

- (3) The overall maturity of the proposed package.

- (4) The Related Work discussion is strong, overall.

Its structure, populated references are pertinent and up to date, while the level of insights is a very good trade-off to make an efficient use of the text real estate.

- (5) The performance of the proposed contribution, despite the relatively competitive field.

** Weaknesses:

- (1) The experimental quantitative evaluation lacks perspective.

This applies throughout and hurts the contributions:
(i) decomposition contribution: Mask-RCNN (He 2017) is now considered a standard baseline rather than a strong state-of-the-art reference. In turn, the near perfect performance of the proposed method on the proposed dataset raises questions regarding the pertinence of the selected scenes to evaluate altogether. Having a more compelling baseline or multiple ones could help the reader accept the proposed contribution at the method level, near perfect. In absence of the aforementioned, it is still difficult to accept as is.

(ii) Object Manipulation: there should be a better justification for a complete lack of a baseline, outside the ablative study as an experimental contribution.


(iii) The contributed dataset: the proof of concept applications to validate its usefullness and quality is strongly tied to (i) above, which makes it slightly problematic as it currently stands.


- (2) How much does it cost? (Conceptual discussion)

There are missing information about typical runtimes, resource usage and specificities, in particular in contrast to the closest related NERF-based derivatives out there.

**Summary Of The Paper:**

The paper proposes a novel formulation based on Neural Radiance Fields (NERFs) to jointly reconstruc, semantically decompose, manipulate and render complex scenes in a unified pipeline.

**Summary Of The Review:**


There are multiple positives to this one. However, the main missing bits hurt all three levels of contributions made by the submitted paper as mentioned above.

---

> ### Author Response · Authors · 2022-11-20
> **Response to Reviewer 1RkG**
>
> **Q2: How much does it cost (Conceptual discussion)? There are missing information about typical runtimes, resource usage and specificities, in particular in contrast to the closest related NERF-based derivatives out there.**
>
> **A2:** As requested, we provide detail information for computation in Section A.7 of Appendix. In particular, we typically train 200K iterations in $\sim$15 hours on each scene ($\sim$0.27s for each iteration) with the batch size of 3072 rays, which uses $\sim$24GB GPU memory. In contrast, the original NeRF (rendering only) needs $\sim$0.22s for each iteration). During the inference of joint decomposition and rendering, our DM-NeRF costs $\sim$9.3s per image with the batch size of 4096 rays using $\sim$5GB GPU memory. For joint decomposition, manipulation and rendering, $\sim$23.4s are required for each image and $\sim$9GB GPU memory is needed when the batch size is set as 4096 rays. To render an image from a novel view, the original NeRF and Point-NeRF need $\sim$7.8s and $\sim$8.2s, respectively. All training and testing are operated on a single Nvidia GeForce RTX 3090 card.
>
> **Q3: Most of it is on point, but there is one missing link between the 3 layers of contributions, as stated above, which currently makes it stlightly of a tough sell without this missing justification or experimental addition. // There are multiple positives to this one. However, the main missing bits hurt all three levels of contributions made by the submitted paper as mentioned above.**
>
> **A3:** As requested, in the revised paper, we add Swin Transformer as an additional baseline for scene decomposition, add Point-NeRF as baseline for object manipulation for comparison. Besides, we also create 5 additional versions of noisy DM-SR dataset to extensively evaluate the robustness of our method. As shown in Tables 1\&2\&3 of the main paper and Figures 9\&10\&11 of Appendix, our method demonstrates superior results for both 3D scene decomposition and object manipulation over all baselines. We believe that these are indeed valuable for this emerging research topic.

---

> ### Author Response · Authors · 2022-11-20
> **Response to Reviewer 1RkG**
>
> **Q1.2 Object Manipulation: there should be a better justification for a complete lack of a baseline, outside the ablative study as an experimental contribution.**
>
> **A1.2:** Thanks for this crucial suggestion. In the revised paper,  we add the very recent Point-NeRF [2] as a baseline via careful adaptations as detailed in Section A.8 of Appendix. In Section 4.4 “3D Object Manipulation", Table 3 shows quantitative results and Figure 11 of Appendix shows qualitative comparison on DM-SR dataset. Our method clearly shows more fine-grained manipulation results, while Point-NeRF produces holes and blurring artifacts after manipulations. Due to the limited time, part of experiments are still running and the missing numbers in Table 3 will be completed within one week.
>
> [2] Point-NeRF: Point-based Neural Radiance Fields, CVPR 2022.
>
> **Table 3: Quantitative results of object manipulation and ablation studies on DM-SR dataset.**
> |                            |           |           |           |           |                            |           |           |           |           |
> |----------------------------|-----------|-----------|-----------|-----------|----------------------------|-----------|-----------|-----------|-----------|
> |       **Translation**      |   PSNR↑   |   SSIM↑   |   LPIPS↓  |   AP$^{0.9}$↑  |        **Rotation**        |   PSNR↑   |   SSIM↑   |   LPIPS↓  |   AP$^{0.9}$↑  |
> | Point-NeRF (on Study)      |   29.85   |   0.933   |   0.128   |     -     | Point-NeRF (on Study)      |   24.51   |   0.918   |   0.140   |     -     |
> | Ours (on Study)            |   32.71   |   0.975   |   0.046   |   94.04   | Ours (on Study)            |   27.94   |   0.961   |   0.055   |   90.10   |
> | Point-NeRF                 |     -     |     -     |     -     |     -     | Point-NeRF                 |     -     |     -     |     -     |     -     |
> | Ours                       | **33.94** | **0.975** | **0.033** | **89.33** | Ours                       | **31.94** | **0.969** | **0.038** | **85.68** |
> | Ablation 1: Ours (w/o $l_{3d}$) |   32.84   |   0.967   |   0.048   |   87.26   | Ablation 1: Ours (w/o $l_{3d}$) |   30.38   |   0.945   |   0.090   |   82.46   |
> | Ablation 2: Ours (w/o VO)  |   33.54   |   0.970   |   0.045   |   86.93   | Ablation 2: Ours (w/o VO)  |   30.57   |   0.953   |   0.076   |   82.43   |
> |          **Scale**         |   PSNR↑   |   SSIM↑   |   LPIPS↓  |   AP$^{0.9}$↑  |          **Joint**         |   PSNR↑   |   SSIM↑   |   LPIPS↓  |   AP$^{0.9}$↑  |
> | Point-NeRF (on Study)      |   26.12   |   0.929   |   0.121   |     -     | Point-NeRF (on Study)      |   21.46   |   0.908   |   0.142   |     -     |
> | Ours (on Study)            |   30.23   |   0.959   |   0.054   |   92.40   | Ours (on Study)            |   27.26   |   0.955   |   0.057   |   86.92   |
> | Point-NeRF                 |     -     |     -     |     -     |     -     | Point-NeRF                 |     -     |     -     |     -     |     -     |
> | Ours                       | **33.40** | **0.971** | **0.037** | **86.05** | Ours                       | **30.65** | **0.965** | **0.045** | **81.70** |
> | Ablation 1: Ours (w/o $l_{3d}$) |   31.84   |   0.959   |   0.062   |   83.31   | Ablation 1: Ours (w/o $l_{3d}$) |   29.95   |   0.947   |   0.088   |   77.36   |
> | Ablation 2: Ours (w/o VO)  |   32.43   |   0.964   |   0.054   |   76.33   | Ablation 2: Ours (w/o VO)  |   29.85   |   0.951   |   0.075   |   74.71   |
> |                            |           |           |           |           |                            |           |           |           |           |
>
> **Q1.3 The Contributed Dataset: the proof of concept applications to validate its usefullness and quality is strongly tied to (i) above, which makes it slightly problematic as it currently stands.**
>
> **A1.3:** Please refer to our response in above **A1.1**.

---

> > ### Author Response · Authors · 2022-11-28
> > **Additional Quantitative Results**
> >
> > To further support our previous reponses, additional quantitative results are presented as follows:
> >
> > **1. Point-NeRF for Object Manipulation (Q1.2)**
> >
> > The complete results of Point-NeRF are updated in Table 4 by training Point-NeRF models in scene-specific fashion on all 8 scenes of the DM-SR dataset and apply the same manipulation information (Translation, Rotation, Scale and Joint) with our DM-SR to its learned point clouds followed by novel view rendering.
> >
> > **Table 4: Quantitative results of object manipulation and ablation studies on DM-SR dataset.**
> > |                            |           |           |           |           |                            |           |           |           |           |
> > |----------------------------|-----------|-----------|-----------|-----------|----------------------------|-----------|-----------|-----------|-----------|
> > |       **Translation**      |   PSNR↑   |   SSIM↑   |   LPIPS↓  |   AP$^{0.9}$↑  |        **Rotation**        |   PSNR↑   |   SSIM↑   |   LPIPS↓  |   AP$^{0.9}$↑  |
> > | Point-NeRF                 |     25.79     |     0.847     |     0.202     |     -     | Point-NeRF                 |     25.21     |     0.818     |     0.203     |     -     |
> > | Ours                       | **33.94** | **0.975** | **0.033** | **89.33** | Ours                       | **31.94** | **0.969** | **0.038** | **85.68** |
> > | Ablation 1: Ours (w/o $l_{3d}$) |   32.84   |   0.967   |   0.048   |   87.26   | Ablation 1: Ours (w/o $l_{3d}$) |   30.38   |   0.945   |   0.090   |   82.46   |
> > | Ablation 2: Ours (w/o VO)  |   33.54   |   0.970   |   0.045   |   86.93   | Ablation 2: Ours (w/o VO)  |   30.57   |   0.953   |   0.076   |   82.43   |
> > |          **Scale**         |   PSNR↑   |   SSIM↑   |   LPIPS↓  |   AP$^{0.9}$↑  |          **Joint**         |   PSNR↑   |   SSIM↑   |   LPIPS↓  |   AP$^{0.9}$↑  |
> > | Point-NeRF                 |     25.83     |     0.848     |     0.202     |     -     | Point-NeRF                 |     23.51     |     0.816     |     0.226     |     -     |
> > | Ours                       | **33.40** | **0.971** | **0.037** | **86.05** | Ours                       | **30.65** | **0.965** | **0.045** | **81.70** |
> > | Ablation 1: Ours (w/o $l_{3d}$) |   31.84   |   0.959   |   0.062   |   83.31   | Ablation 1: Ours (w/o $l_{3d}$) |   29.95   |   0.947   |   0.088   |   77.36   |
> > | Ablation 2: Ours (w/o VO)  |   32.43   |   0.964   |   0.054   |   76.33   | Ablation 2: Ours (w/o VO)  |   29.85   |   0.951   |   0.075   |   74.71   |
> > |                            |           |           |           |           |                            |           |           |           |           |
> >
> > **2. Inaccurate Supervision Signals (Q3)**
> >
> > We use the instance masks estimated by Mask-RCNN as the supervision signals when training our DM-NeRF. As shown in Figure 10 of Appendix and Table 5, we can see that even though the 2D labels are inaccurate for training, our method still achieves excellent object decomposition results. This result is consistent with that of noisy 2D labels in Section 4.3, showing the remarkable robustness of our method and the applicability in real world.
> >
> > **Table 5: Quantitative object decomposition AP$^{0.75}$ scores of our method trained on inaccurate 2D labels (estimated by MaskRCNN).**
> > |         |          |         |         |         |           |       |       |        |         |
> > |---------|----------|---------|---------|---------|-----------|-------|-------|--------|---------|
> > | Scenes  | Bathroom | Bedroom | Dinning | Kitchen | Reception | Rest  | Study | Office | Average |
> > | Ours_MR | 95.30    | 97.08   | 95.69   | 94.72   | 99.62     | 98.58 | 97.72 | 98.08  | 97.10   |
> > |         |          |         |         |         |           |       |       |        |         |

---

> ### Author Response · Authors · 2022-11-20
> **Response to Reviewer 1RkG**
>
> We appreciate the reviewer's insightful comments and address all concerns below. Would you also check out our revised main paper (highlighted in yellow), and supplied materials (new appendix + new video demo + code) anonymous at [https://github.com/DM-NeRF-ICLR-Rebuttal/DM-NeRF](https://github.com/DM-NeRF-ICLR-Rebuttal/DM-NeRF).
>
> **Q1: The experimental quantitative evaluation lacks perspective. This applies throughout and hurts the contributions:**
>
> **Q1.1 Decomposition Contribution: Mask-RCNN is now considered a standard baseline rather than a strong state-of-the-art reference. In turn, the near perfect performance of the proposed method on the proposed dataset raises questions regarding the pertinence of the selected scenes to evaluate altogether. Having a more compelling baseline or multiple ones could help the reader accept the proposed contribution at the method level, near perfect. In absence of the aforementioned, it is still difficult to accept as is.**
>
> **A1.1:** Thank you for the great advice on baselines and evaluation of DM-SR dataset. In the revised paper, we add the very strong Swin Transformer [1] as an additional baseline for scene decomposition. As also suggested by Reviewer BSjr, in Section 4.3, we conduct extra extensive experiments on 5 variants of noisy DM-SR dataset to evaluate the robustness of our method regarding noisy 2D object labels during training. Tables 1\&2 present quantitative results.
>
> We can see that our method is clearly more accurate and robust than both Swin Transformer and MaskRCNN for 3D scene decomposition. Figure 9 of the Appendix shows qualitative results of our models trained with different amounts of noisy 2D labels. Notably, given 80\% of 2D object labels with noise on our DM-SR dataset, both Swin Transformer and MaskRCNN fail catastrophically (AP$^{0.75}<$4\%), while our method still achieves AP$^{0.75}$ = 74.08\% in object segmentation results.
>
> Regarding our DM-SR dataset, we respectfully argue that its core value lies in the availability of ground truth data given different types of 3D object manipulation, thanks to the blender engine. Nevertheless, the synthetic scene appearance is indeed less challenging than real-world dataset such as ScanNet. We hope our work could inspire more diverse and realistic datasets in the future for the community which is still in its infancy.
>
> [1] Swin Transformer: Hierarchical Vision Transformer using Shifted Windows, ICCV 2021.
>
> **Table 1: Quantitative object decomposition AP$^{0.75}$ scores on DM-SR dataset.**
> |||||
> |-:|:-:|:-:|:-:|
> ||   MR  | SiwnT | **Ours** |
> |  Bathroom | 93.81 | 98.89 |   100.0  |
> |   Bedroom | 97.92 | 98.85 |   100.0  |
> |   Dinning | 98.85 | 97.81 |   99.66  |
> |   Kitchen | 92.06 | 98.81 |   100.0  |
> | Reception | 98.81 | 95.74 |   100.0  |
> |      Rest | 98.89 | 94.50 |   99.89  |
> |     Study | 96.86 | 97.88 |   98.86  |
> |    Office | 97.83 | 96.87 |   100.0  |
> |   Average | 96.87 | 97.42 |   **99.80**  |
> |||||
>
> **Table 2: Quantitative object decomposition AP$^{0.75}$ scores on noisy DM-SR dataset.**
> |||||||||||||||||
> |-:|:-:|:-:|:-:|:-:|:-:|:-:|:-:|:-:|:-|:-:|:-:|:-:|:-:|:-:|:-:|
> | **Noisy Labels** | **10%** |       |           | **50%** |       |           | **70%** |       |           | **80%** |       |           | **90%** |       |           |
> ||    MR   | SiwnT |  **Ours** |    MR   | SiwnT |  **Ours** |    MR   | SiwnT |  **Ours** |    MR   | SiwnT |  **Ours** |    MR   | SiwnT |  **Ours** |
> |         Bathroom |  98.93  | 98.96 |   99.81   |  54.52  | 62.94 |   99.63   |   1.86  |  7.71 |   99.02   |   1.89  |  7.78 |   58.09   |   1.79  |  7.11 |    9.69   |
> |          Bedroom |  98.61  | 98.85 |   100.0   |  75.41  | 92.94 |   100.0   |   3.00  |  3.96 |   100.00  |   2.96  |  3.18 |   82.83   |   2.95  |  2.75 |    4.25   |
> |          Dinning |  95.57  | 97.88 |   98.41   |  46.52  | 43.50 |   85.48   |   1.40  |  1.05 |   81.91   |   1.51  |  1.09 |   63.50   |   1.36  |  0.96 |   14.44   |
> |          Kitchen |  98.82  | 98.81 |   100.0   |  84.93  | 93.60 |   100.0   |   4.09  |  8.42 |   100.0   |   4.02  |  5.04 |   51.87   |   4.04  |  4.69 |    1.80   |
> |        Reception |  80.96  | 91.00 |   100.0   |  32.78  | 42.11 |   100.0   |   1.03  |  6.27 |   100.0   |   0.73  |  1.96 |   100.0   |   0.22  |  1.64 |   37.63   |
> |     Rest |  93.58  | 94.50 |   99.64   |  52.75  | 51.99 |   99.32   |   1.63  |  1.57 |   99.33   |   2.46  |  2.02 |   66.74   |   1.63  |  1.69 |   11.11   |
> |    Study |  93.07  | 97.94 |   98.58   |  49.03  | 60.04 |   97.97   |   1.44  |  6.24 |   98.03   |   1.50  |  6.06 |   72.62   |   1.40  |  5.48 |   40.72   |
> |    Office |  95.09  | 97.00 |   100.0   |  66.28  | 69.11 |   100.0   |   4.13  |  2.96 |   100.0   |   2.61  |  3.94 |   97.00   |   2.66  |  2.48 |    0.00   |
> |  Average |  94.33  | 96.87 | **99.56** |  57.78  | 64.53 | **97.80** |   2.32  |  4.77 | **97.29** |   2.21  |  3.88 | **74.08** |   2.00  |  3.35 | **14.96** |
> |||||||||||||||||

---

> ### Author Response · Authors · 2022-12-14
> **Waiting for feedback**
>
> Dear reviewer 1RkG,
>
> Since the discussion period is closing very soon, we are still waiting for your thoughts on our rebuttal materals and the improved manuscirpt.
>
> Regarding all your concerns about 1) more baselines for object decomposition and manipulation, 2) our dataset, and 3) computation cost, etc, we believe they are all clearly addressed and also highlighted by yellow color in the main paper and the new appendix.
>
> We are grateful if you could share your further feedback. Thank you for your time.
>
> Regards,
> Authors

---

### Official Review · Reviewer_BSjr · 2022-10-25

**Confidence:** 3
**Correctness:** 3
**Technical Novelty And Significance:** 3
**Empirical Novelty And Significance:** 3
**Recommendation:** 6

**Clarity, Quality, Novelty And Reproducibility:**

The paper is mostly clearly written, though the source code is not provided. This work is an extension of NeRF, which also predicts an object code for segmentation. The major novelties are segmentation supervision and the inverse query algorithm for object manipulation.

**Strength And Weaknesses:**

- The task studied in this work is important. As the authors mentioned, many cutting-edge tasks may require obtaining and manipulating individual 3D objects in a 3D scene.
- The object code supervision proposed by the authors is natural and generates good results. As the authors mentioned, supervising the object code prediction is challenging as the ordering of the objects is not determined. The multiple views of 2D object code with the Hungarian algorithm is a good solution to it (idea originated from 3D-BoNet), and it produces better results than Mask-RCNN.
- The object manipulator produced reasonable results but has space for improvement. Empirically, the manipulator produced artifacts (Figure 7, first line). At the end of the video “DM-NeRF_2D.mp4”, when the table is being rotated, they are some very unrealistic artifacts as well as unrealistic light.

**Summary Of The Paper:**

This submission proposed an extension of NeRF called DM-NeRF, which can segment individual objects and manipulate them (translation, rotation, deformation, etc.). To supervise object segmentation, the authors use their algorithm to generate multiple 2D object code predictions and match them with ground truth object labels using the Hungarian algorithm. To manipulate an object, they perform the inverse operation on sample points along a ray. The empirical results show high-quality novel view synthesis and better decomposition than Mask-RCNN.

**Summary Of The Review:**

The authors proposed to extend NeRF object instance segmentation, providing a way to supervise the object code prediction which produced good results (better than Mask-RCNN). They also proposed an object manipulator which supports translating/rotating/scaling an object but generates artifacts and did not consider lighting conditions, which is my major concern.

---

> ### Author Response · Authors · 2022-11-20
> **Response to Reviewer BSjr**
>
> We appreciate the reviewer's very encouraging comments and make responses below. Would you also check out our revised main paper (highlighted in yellow), and supplied materials (new appendix + new video demo + code) anonymous at [https://github.com/DM-NeRF-ICLR-Rebuttal/DM-NeRF](https://github.com/DM-NeRF-ICLR-Rebuttal/DM-NeRF).
>
> **Q1: The object manipulator produced reasonable results but has space for improvement. Empirically, the manipulator produced artifacts (Figure 7, first line). At the end of the video "DM-NeRF\_2D.mp4", when the table is being rotated, they are some very unrealistic artifacts as well as unrealistic light.**
>
> **A1:** Thank you for pointing out the flaws in our previous manipulation results, as also mentioned by Reviewer tL8f. Fundamentally, such artifacts are likely caused by the manipulation of originally occluded objects or parts with inaccurate learned radiance fields. However, they can be largely repaired by applying simple heuristics such as continuity of surfaces or using better neural backbones in the future.
>
> We explicitly add a discussion of limitation in the last paragraph of revised main paper. In addition, we also show in (Figure 12 + Artifacts\_Repairing.mp4) of Appendix that those artifacts can be repaired by simply using a neighbourhood voting scheme.
>
> **Q2: The paper is mostly clearly written, though the source code is not provided. This work is an extension of NeRF, which also predicts an object code for segmentation. The major novelties are segmentation supervision and the inverse query algorithm for object manipulation.**
>
> **A2:** Thank you for appreciating the novelties and writing of our paper. For reproducing all results and benefiting the community, our full code will be made public and is currently anonymous at [https://github.com/DM-NeRF-ICLR-Rebuttal/DM-NeRF](https://github.com/DM-NeRF-ICLR-Rebuttal/DM-NeRF).
>
> **Q3: The authors proposed to extend NeRF object instance segmentation, providing a way to supervise the object code prediction which produced good results (better than Mask-RCNN). They also proposed an object manipulator which supports translating/rotating/scaling an object but generates artifacts and did not consider lighting conditions, which is my major concern.**
>
> **A3:** We agree that editing lighting conditions is desirable but not considered in this paper. In fact, manipulation of lighting conditions involves sophisticated designs to decouple lighting factors in rendering equations. This is another exciting yet non-trivial research problem, and we leave it for future exploration.

---

### Official Review · Reviewer_tL8f · 2022-10-25

**Confidence:** 4
**Correctness:** 4
**Technical Novelty And Significance:** 3
**Empirical Novelty And Significance:** 3
**Recommendation:** 8

**Clarity, Quality, Novelty And Reproducibility:**

The paper is easy to read. However there are a few typos, just to name a few:
Section 1, 3rd paragraph: simple amodel -> simple model
Section 1, 4th paragraph: with a ability to clearly -> with an ability to clearly
Section 3, 1st paragraph: this villina NeRF -> this vanilla NeRF
Section 3, 1st paragraph: together with a object -> together with an object


**Strength And Weaknesses:**

+ The idea is novel. The design of the object code and the loss functions is simple and elegant.
+ The authors have done extensive experiments on a few different datasets and showed superior results comparing with MaskRCNN
+ This paper is a resubmission of an ECCV 2022 submission. The comments in the review and rebuttal have been addressed quite well in this submission.

- The authors mainly compared their approach with only one existing approach (MaskRCNN). However, I do agree with the authors' explanation in section 4.2 that most relevant work focuses on similar but different problems thus it is hard to do a fair comparison with these works.
- In all manipulation results, the proposed method still has a ghost boundary along the model pre-manipulation. Can you discuss this limitation?


**Summary Of The Paper:**

This paper extends on the power NeRF, by introducing an additional vector to present the object code for each 3D point. The authors designed a few loss functions and showed that with these loss functions the object code for the 3D points can be directly optimized from 2D image annotations. The authors then used the learnt object code to translate/rotate/scale the objects and showed promising experimental results.

**Summary Of The Review:**

This paper presents a good idea and executes it well. The design of the object code vector and the loss functions is neat. The authors show superior accuracy compared with MaskRCNN. They further used this to demonstrate an application of manipulating the objects. This paper is a resubmission of an ECCV 2022 submission. The authors addressed lots of comments from the previous review cycle, and have made it a good submission for ICLR.

---

> ### Author Response · Authors · 2022-11-20
> **Response to Reviewer tL8f**
>
>
> **Table 3: Quantitative results of object manipulation and ablation studies on DM-SR dataset.**
> |                            |           |           |           |           |                            |           |           |           |           |
> |----------------------------|-----------|-----------|-----------|-----------|----------------------------|-----------|-----------|-----------|-----------|
> |       **Translation**      |   PSNR↑   |   SSIM↑   |   LPIPS↓  |   AP$^{0.9}$↑  |        **Rotation**        |   PSNR↑   |   SSIM↑   |   LPIPS↓  |   AP$^{0.9}$↑  |
> | Point-NeRF (on Study)      |   29.85   |   0.933   |   0.128   |     -     | Point-NeRF (on Study)      |   24.51   |   0.918   |   0.140   |     -     |
> | Ours (on Study)            |   32.71   |   0.975   |   0.046   |   94.04   | Ours (on Study)            |   27.94   |   0.961   |   0.055   |   90.10   |
> | Point-NeRF                 |     -     |     -     |     -     |     -     | Point-NeRF                 |     -     |     -     |     -     |     -     |
> | Ours                       | **33.94** | **0.975** | **0.033** | **89.33** | Ours                       | **31.94** | **0.969** | **0.038** | **85.68** |
> | Ablation 1: Ours (w/o $l_{3d}$) |   32.84   |   0.967   |   0.048   |   87.26   | Ablation 1: Ours (w/o $l_{3d}$) |   30.38   |   0.945   |   0.090   |   82.46   |
> | Ablation 2: Ours (w/o VO)  |   33.54   |   0.970   |   0.045   |   86.93   | Ablation 2: Ours (w/o VO)  |   30.57   |   0.953   |   0.076   |   82.43   |
> |          **Scale**         |   PSNR↑   |   SSIM↑   |   LPIPS↓  |   AP$^{0.9}$↑  |          **Joint**         |   PSNR↑   |   SSIM↑   |   LPIPS↓  |   AP$^{0.9}$↑  |
> | Point-NeRF (on Study)      |   26.12   |   0.929   |   0.121   |     -     | Point-NeRF (on Study)      |   21.46   |   0.908   |   0.142   |     -     |
> | Ours (on Study)            |   30.23   |   0.959   |   0.054   |   92.40   | Ours (on Study)            |   27.26   |   0.955   |   0.057   |   86.92   |
> | Point-NeRF                 |     -     |     -     |     -     |     -     | Point-NeRF                 |     -     |     -     |     -     |     -     |
> | Ours                       | **33.40** | **0.971** | **0.037** | **86.05** | Ours                       | **30.65** | **0.965** | **0.045** | **81.70** |
> | Ablation 1: Ours (w/o $l_{3d}$) |   31.84   |   0.959   |   0.062   |   83.31   | Ablation 1: Ours (w/o $l_{3d}$) |   29.95   |   0.947   |   0.088   |   77.36   |
> | Ablation 2: Ours (w/o VO)  |   32.43   |   0.964   |   0.054   |   76.33   | Ablation 2: Ours (w/o VO)  |   29.85   |   0.951   |   0.075   |   74.71   |
> |                            |           |           |           |           |                            |           |           |           |           |
>
> **Q2: In all manipulation results, the proposed method still has a ghost boundary along the model pre-manipulation. Can you discuss this limitation?**
>
> **A2:** Thank you for pointing out the flaws in our previous manipulation results, as also mentioned by Reviewer BSjr. Fundamentally, such artifacts are likely caused by the manipulation of originally occluded objects or parts with inaccurate learned radiance fields. However, they can be largely repaired by applying simple heuristics such as continuity of surfaces or using better neural backbones in the future.
>
> As suggested, we explicitly add a discussion of limitation in the last paragraph of revised main paper. In addition, we also show in (Figure 12 + Artifacts\_Repairing.mp4) of Appendix that those artifacts can be largely repaired by simply using a neighbourhood voting scheme.
>
> **Q3: The paper is easy to read. However there are a few typos, just to name a few: Section 1, 3rd paragraph: simple amodel $\rightarrow$ simple model Section 1, 4th paragraph: with a ability to clearly $\rightarrow$ with an ability to clearly Section 3, 1st paragraph: this villina NeRF $\rightarrow$ this vanilla NeRF Section 3, 1st paragraph: together with a object $\rightarrow$ together with an object.**
>
> **A3:** We really appreciate your time to carefully identify these typos, which are all fixed in the revised main paper.

---

> ### Author Response · Authors · 2022-11-20
> **Response to Reviewer tL8f**
>
> We appreciate the reviewer's very positive comments and make responses as follows. Would you also check out our revised main paper (highlighted in yellow), and supplied materials (new appendix + new video demo + code) anonymous at [https://github.com/DM-NeRF-ICLR-Rebuttal/DM-NeRF](https://github.com/DM-NeRF-ICLR-Rebuttal/DM-NeRF).
>
> **Q1: The authors mainly compared their approach with only one existing approach (MaskRCNN). However, I do agree with the authors' explanation in section 4.2 that most relevant work focuses on similar but different problems thus it is hard to do a fair comparison with these works.**
>
> **A1:** Thank you for your agreement with the lack of comparable baselines in literature. In the revised paper, as also requested by Reviewer 1RkG, we add Swin Transformer [1] as an additional baseline for object segmentation. In Section 4.3 "3D Scene Decomposition", Tables 1\&2 present quantitative results on the original DM-SR dataset and 5 versions of noisy DM-SR dataset. We can see that our method is clearly more accurate and robust than both Swin Transformer and MaskRCNN for 3D scene decomposition.
>
> For 3D object manipulation, as also requested by Reviwer 1RkG, we also add the very recent Point-NeRF [2] as a baseline via careful adaptations. In Section 4.4 "3D Object Manipulation", Table 3 shows quantitative results and Figure 11 of Appendix shows qualitative comparison on DM-SR dataset. Our method clearly shows more fine-grained manipulation results, while Point-NeRF produces holes and blurring artifacts after manipulations.
>
> [1] Swin Transformer: Hierarchical Vision Transformer using Shifted Windows, ICCV 2021.
>
> [2] Point-NeRF: Point-based Neural Radiance Fields, CVPR 2022.
>
> **Table 1: Quantitative object decomposition AP$^{0.75}$ scores on DM-SR dataset.**
> |||||
> |-:|:-:|:-:|:-:|
> ||   MR  | SiwnT | **Ours** |
> |  Bathroom | 93.81 | 98.89 |   100.0  |
> |   Bedroom | 97.92 | 98.85 |   100.0  |
> |   Dinning | 98.85 | 97.81 |   99.66  |
> |   Kitchen | 92.06 | 98.81 |   100.0  |
> | Reception | 98.81 | 95.74 |   100.0  |
> |      Rest | 98.89 | 94.50 |   99.89  |
> |     Study | 96.86 | 97.88 |   98.86  |
> |    Office | 97.83 | 96.87 |   100.0  |
> |   Average | 96.87 | 97.42 |   **99.80**  |
> |||||
>
> **Table 2: Quantitative object decomposition AP$^{0.75}$ scores on noisy DM-SR dataset.**
> |||||||||||||||||
> |-:|:-:|:-:|:-:|:-:|:-:|:-:|:-:|:-:|:-|:-:|:-:|:-:|:-:|:-:|:-:|
> | **Noisy Labels** | **10%** |       |           | **50%** |       |           | **70%** |       |           | **80%** |       |           | **90%** |       |           |
> ||    MR   | SiwnT |  **Ours** |    MR   | SiwnT |  **Ours** |    MR   | SiwnT |  **Ours** |    MR   | SiwnT |  **Ours** |    MR   | SiwnT |  **Ours** |
> |         Bathroom |  98.93  | 98.96 |   99.81   |  54.52  | 62.94 |   99.63   |   1.86  |  7.71 |   99.02   |   1.89  |  7.78 |   58.09   |   1.79  |  7.11 |    9.69   |
> |          Bedroom |  98.61  | 98.85 |   100.0   |  75.41  | 92.94 |   100.0   |   3.00  |  3.96 |   100.00  |   2.96  |  3.18 |   82.83   |   2.95  |  2.75 |    4.25   |
> |          Dinning |  95.57  | 97.88 |   98.41   |  46.52  | 43.50 |   85.48   |   1.40  |  1.05 |   81.91   |   1.51  |  1.09 |   63.50   |   1.36  |  0.96 |   14.44   |
> |          Kitchen |  98.82  | 98.81 |   100.0   |  84.93  | 93.60 |   100.0   |   4.09  |  8.42 |   100.0   |   4.02  |  5.04 |   51.87   |   4.04  |  4.69 |    1.80   |
> |        Reception |  80.96  | 91.00 |   100.0   |  32.78  | 42.11 |   100.0   |   1.03  |  6.27 |   100.0   |   0.73  |  1.96 |   100.0   |   0.22  |  1.64 |   37.63   |
> |     Rest |  93.58  | 94.50 |   99.64   |  52.75  | 51.99 |   99.32   |   1.63  |  1.57 |   99.33   |   2.46  |  2.02 |   66.74   |   1.63  |  1.69 |   11.11   |
> |    Study |  93.07  | 97.94 |   98.58   |  49.03  | 60.04 |   97.97   |   1.44  |  6.24 |   98.03   |   1.50  |  6.06 |   72.62   |   1.40  |  5.48 |   40.72   |
> |    Office |  95.09  | 97.00 |   100.0   |  66.28  | 69.11 |   100.0   |   4.13  |  2.96 |   100.0   |   2.61  |  3.94 |   97.00   |   2.66  |  2.48 |    0.00   |
> |  Average |  94.33  | 96.87 | **99.56** |  57.78  | 64.53 | **97.80** |   2.32  |  4.77 | **97.29** |   2.21  |  3.88 | **74.08** |   2.00  |  3.35 | **14.96** |
> |||||||||||||||||

---

### Author Response · Authors · 2022-11-20
**Overall Response**

We appreciate all valuable comments. After carefully improving the quality of our submission, we present here a revised main paper (highlighted in yellow), and supplied materials (new appendix + new video demo + code) anonymous at [https://github.com/DM-NeRF-ICLR-Rebuttal/DM-NeRF](https://github.com/DM-NeRF-ICLR-Rebuttal/DM-NeRF).

Changes in the main paper have been highlighted and include:

**1. Clarification of our figures, equations and loss functions**

**2. Additional baseline (Swin Transformer) for object segmentation**

**3. Additional baseline (Point-NeRF) for object manipulation**

**4. Additional evaluation on the noisy and inaccurate DM-SR dataset**

**5. Discussion on the limitation of our method**

Specifically, additional experiments and analysis to address the reviewers' concerns are collected into the supplementary materials at [https://github.com/DM-NeRF-ICLR-Rebuttal/DM-NeRF](https://github.com/DM-NeRF-ICLR-Rebuttal/DM-NeRF) and include:

**1. Appendix\_Rebuttal.pdf** (Pages 13-16)
> -  **A.1:** Qualitative Results of 3D Scene Decomposition and Manipulation
> -  **A.2:** Qualitative Results of Our Method Trained on Noisy 2D Labels
> -  **A.3:** Qualitative Results of Our Method Trained on Inaccurate 2D Labels (Estimated by MaskRCNN)
> -  **A.4:** Qualitative Results of Our Method and Point-NeRF for Object Manipulations
> -  **A.5:** Repairing Artifacts
> -  **A.6:** Panoptic Segmentation
> -  **A.7:** Computation
> -  **A.8:** Adaptation of Point-NeRF for Object Manipulation

**2. Artifacts\_Repairing.mp4**

**3. DM-NeRF-Code**

---

### Decision · Program_Chairs · 2023-01-20

**Decision:**

Accept: poster

**Justification For Why Not Higher Score:**

See the Summary of AC-reviewer meeting for the debates during the discussion phase. There are some issues caused by the author side in reviewing the work.

**Justification For Why Not Lower Score:**

The work provides a value idea with a good amount of experiments.

**Metareview: Summary, Strengths And Weaknesses:**

The paper studies the structural decomposition of 3D objects from 2D views.

The idea is novel; however, at the submission time, some critical experiments are missing. Therefore, the paper received mixed scores. The authors provided a strong rebuttal, including experiment results submitted a bit behind the deadline of Stage 1 of the discussion period. After a few rounds of discussions among reviewers and the area chair, the committee recommends to accept the submission.ted.

**Note From Pc:**

if the above contains the word "oral" or "spotlight" please see: "oral" presentation means -> notable-top-5% and "spotlight" means -> notable-top-25%. As stated in our emails, we are disassociating presentation type from AC recommendations

**Summary Of Ac-Reviewer Meeting:**

There are extensive discussions on whether ICLR should accept the submission.

The original experiment results are already good, but a bit insufficient in some aspects. Based on the results during the discussion phase (even the first rebuttal message), the results are strong enough to be accepted.

One issue is that the rebuttals with extensive experiment results are submitted a bit late (two days after the deadline of Stage 1), leaving no time for the reviewers to discuss with the authors. Somehow there is a high score received at the beginning (one score of 8) and two scores positive (two 6), which makes the committee feel that the overall view to the work is quite positive, and there is a core idea valuable to the community. And, as a paper on the OpenReview platform, the rebuttals together with new experiment results will anyway be visible to the public. After a few rounds of discussions (including a virtual session), the committee leans toward accepting this work.

The results and discussions in the rebuttal should be added to the paper at camera ready, as they provide importance evidences in the discussion phase.